# IT TAKES TWO: YOUR GRPO IS SECRETLY DPO

## ABSTRACT

Group Relative Policy Optimization (GRPO) is a prominent reinforcement learning algorithm for post-training Large Language Models (LLMs). It is commonly believed that GRPO necessitates a large group size to ensure the advantage estimate via appropriate statistical estimation, which incurs substantial computational overhead. In this work, we challenge this assumption by reframing GRPO as a form of contrastive learning, where the construction of contrastive examples is more essential. This not only reveals a fundamental connection to Direct Preference Optimization (DPO), but also motivates our investigation on the minimal two-rollout case of GRPO (2-GRPO) — a configuration previously deemed infeasible. We provide a rigorous theoretical analysis to validate 2-GRPO and demonstrate empirically that it achieves performance on par with 16-GRPO, despite using only $1/8$ of the rollouts and reducing training time by over 70%.

## 1 INTRODUCTION

Reinforcement Learning (RL) is now a central paradigm for post-training Large Language Models (LLMs), aligning preference through RL with Human Feedback (RLHF) (Ouyang et al., 2022) and incentivizing reasoning capability through RL with Verifiable Rewards (RLVR) (Shao et al., 2024; DeepSeek-AI, 2025). Among recent advances, *Group Relative Policy Optimization* (GRPO) (Shao et al., 2024; DeepSeek-AI, 2025), has emerged as a powerful variant of Proximal Policy Optimization (PPO) (Schulman et al., 2017). In contrast to PPO, which stabilizes policy optimization using an additional value network, GRPO samples multiple responses (rollouts) for each prompt and normalizes their rewards within the group to compute the corresponding advantages for optimization stabilization. GRPO achieves strong performance on a variety of reasoning tasks while reducing computational costs by eliminating the need for value networks.

Due to its intra-group normalization mechanism, it is commonly believed that GRPO necessitates a larger group size—i.e., more rollouts per prompt—to ensure reliable and stable normalization. However, generating a large number of rollouts per prompt leads to substantial computational and time costs. This raises a natural question: can we reduce the group size to improve computational efficiency without significantly compromising performance?

In this work, we revisit GRPO from the perspective of contrastive learning (Chopra et al., 2005; Wang & Isola, 2020; Chen et al., 2020; He et al., 2020; Wu et al., 2024). This perspective enables us to draw the underlying connection GRPO and Direct Preference Optimization (DPO) (Rafailov et al., 2023) – a prominent offline RL alignment algorithm for human feedback – from the gradients of their objectives. We demonstrate that, in essence, GRPO performs "direct preference optimization," adapted for the online RL setting and leveraging additional positive and negative examples for more precise gradient estimation. Furthermore, this perspective motivates our investigation on the minimal case of GRPO with only two rollouts per prompt (denoted as 2-GRPO), which was viewed as inappropriate from the perspective of reward normalization.

Theoretically, we provide an in-depth analysis of why GRPO can work with a group size of two: (i) 2-GRPO preserves an implicit form of advantage estimation, allowing proper credit assignments; (ii) the potential increase in gradient variance can be mitigated by a larger batch size; and (iii) 2-GRPO does not yield fewer valid signals compared to its large-group counterpart even for challenging questions. Empirically, we demonstrate that 2-GRPO achieves comparable performance to standard GRPO on challenging mathematical benchmarks, while substantially reducing computational overhead and training time.

Our findings challenge the prevailing belief that large group sizes are necessary for the efficacy of GRPO stemming from the perspective of reward normalization. By demonstrating that 2-GRPO is a competitive and substantially more efficient alternative, we offer a new direction for designing resource-efficient RL algorithms for LLM post-training.

## 2 PRELIMINARY

Our work focuses on RL-based post-training of LLMs to improve their reasoning capabilities, with particular emphasis on the setting of verifiable rewards (RLVR) – i.e., the responses can be verified as correct or incorrect.

We denote the policy model as $\pi_\theta$, i.e., the LLM with parameters $\theta$. Let $\mathcal{Q}$ be the set of prompts, each consisting of a question and any necessary instructions[1]. Given an input prompt $q \in \mathcal{Q}$, the model generates a response $o_i = (o_{i,1}, \ldots, o_{i,T})$, where $o_{i,t}$ is the token generated at step $t \in [0, T]$ and $o_{i,<t}$ denotes the sequence of preceding tokens. A trajectory $\tau \in \mathcal{T}$ is defined as a pair consisting of a prompt $q \in \mathcal{Q}$ and its corresponding LLM-generated response sequence $o$, i.e., $\tau = (q, o)$. In RL-based post-training, the reward function is typically defined at the trajectory level, i.e., $r : \mathcal{T} \to \mathbb{R}$. The learning objective is to maximize the expected reward over the space of trajectories:

$$\mathcal{J}(\theta) = \mathbb{E}_{q \sim \mathcal{Q}} \mathbb{E}_{o \sim \pi_\theta(\cdot|q)}[r(\tau)] . \tag{1}$$

**Vanilla Policy Gradient (VPG) (Williams, 1992):** VPG (a.k.a. REINFORCE) aims to maximize the reward with gradient ascent:

$$\nabla_\theta \mathcal{J}(\theta) = \mathbb{E}_{[q \sim \mathcal{Q}]} \mathbb{E}_{[o_i \sim \pi_\theta(\cdot|q)]} \left[ r_i \sum_{t=0}^{|o_i|} \nabla_\theta \log \pi_\theta(o_{i,t}|o_{i,<t}, q) \right] . \tag{2}$$

where $r_i$ is the reward of $(q, o_i)$.

**Proximity Policy Optimization (PPO) (Schulman et al., 2017):** VPG usually suffers from high gradient variance and training instability (Schulman et al., 2015). To mitigate the issues, especially for online off-policy setting, PPO introduces importance sampling with clipping, and advantage estimation with value model:

$$\mathcal{J}_{\text{PPO}}(\theta) = \mathbb{E}_{\left[ \substack{q \sim \mathcal{Q} \\ o_i \sim \pi_{\theta_{\text{old}}}} \right]} \frac{1}{T} \sum_{t=1}^{T} \min \left[ \frac{\pi_\theta(o_{i,t}|o_{i,<t}, q)}{\pi_{\theta_{\text{old}}}(o_{i,t}|o_{i,<t}, q)} A_{i,t}, \text{clip}\left( \frac{\pi_\theta(o_{i,t}|o_{i,<t}, q)}{\pi_{\theta_{\text{old}}}(o_{i,t}|o_{i,<t}, q)}, 1 - \epsilon, 1 + \epsilon \right) A_{i,t} \right] ,$$

$$\tag{3}$$

where $\pi_{\theta_{\text{old}}}$ is the old policy which generates the responses, $\pi_\theta$ is the current policy to update, $\epsilon$ is a hyperparameter for clipping, and $A_{i,t}$ is the advantage, which is computed as $r_i$ minus a value baseline. The value baseline is provided by a value model, which is typically parameterized as a LLM.

**Direct Preference Optimization (DPO) (Rafailov et al., 2023):** DPO is proposed for RLHF, which is usually trained with offline human-annotated preference data $(q, o_+, o_-) \sim \mathcal{D}_{\text{DPO}}$. The loss function of DPO is

$$\mathcal{L}_{\text{DPO}} = -\mathbb{E}_{[(q, o_+, o_-) \sim \mathcal{D}_{\text{DPO}}]} \left[ \log \sigma \left( \beta \log \frac{\pi_\theta(o_+|q)}{\pi_{\text{ref}}(o_+|q)} - \beta \log \frac{\pi_\theta(o_-|q)}{\pi_{\text{ref}}(o_-|q)} \right) \right] , \tag{4}$$

where $o_+$ and $o_-$ denote a preferred (positive) and a dispreferred (negative) response, respectively; $\sigma$ is the logistic function; $\beta$ is a parameter controlling the deviation from the base reference policy $\pi_{\text{ref}}$.

**Group Relative Policy Optimization (GRPO) (Shao et al., 2024):** GRPO – the RL algorithm behind the success of DeepSeek-R1 (DeepSeek-AI, 2025) – has become one of the most widely used RL algorithms for LLM post-training. Instead of maintaining a value network like PPO, GRPO

---

[1]Throughout this paper, we use the terms "prompt" and "question" interchangeably.

generates a group of $G$ trajectories for each prompt (usually referred to as rollouts), and normalizes the corresponding rewards within each group to estimate the advantages:

$$\mathcal{J}_{\text{GRPO}}(\theta) = \mathbb{E}_{\left[q \sim \mathcal{Q}, \{o_i\}_{i=1}^{G} \sim \pi_{\theta_{\text{old}}}\right]}$$

$$\frac{1}{G} \sum_{i=1}^{G} \frac{1}{|o_i|} \sum_{t=1}^{|o_i|} \min\left[ \frac{\pi_\theta(o_{i,t}|o_{i,<t},q)}{\pi_{\theta_{\text{old}}}(o_{i,t}|o_{i,<t},q)} A_{i,t}, \text{clip}\left( \frac{\pi_\theta(o_{i,t}|o_{i,<t},q)}{\pi_{\theta_{\text{old}}}(o_{i,t}|o_{i,<t},q)}, 1-\epsilon, 1+\epsilon \right) A_{i,t} \right] , \quad (5)$$

where $G \geq 2$, $o_i$ denotes the $i$-th trajectory, and $A_{i,t}$ denotes the corresponding advantage.

Though expressed as token-level advantages, GRPO actually introduces sequence-level advantages, given by the intra-group normalization as follows:

$$A_{i,t} = \frac{r_i - \text{mean}(\mathbf{r})}{\text{std}(\mathbf{r}) + \epsilon}, \quad (6)$$

where $r_i \in \mathbf{r}$ is the reward of the rollout, and $\epsilon$ is a small constant added to avoid division by zero. If all trajectories within a group receive identical rewards (i.e., all correct or all incorrect), then $A_{i,t} = 0$ for all $i, t$, resulting in zero gradients during optimization. In practice, $G$ is often set to a relatively large value (e.g., 16) to guarantee effective normalization.

## 3 Bridging GRPO and DPO with Contrastive Learning

At first glance, the objectives of GRPO and DPO appear distinct. In this section, we show that, from the standpoint of contrastive learning, the gradients of GRPO and DPO actually share the same underlying form. More importantly, both GRPO and DPO estimate an expectation invariant to group size. This discovery allows us to reduce the group size in GRPO, significantly reducing generation time and remaining the unbiasedness of estimation.

### 3.1 Contrastive Loss for Sequences

A variety of contrastive loss functions (Chopra et al., 2005) have been proposed in the literature—ranging from 1-vs-1 (one positive sample and one negative sample) (Rendle et al., 2009), to 1-vs-$N$ (Oord et al., 2018), and $N$-vs-$M$ (Frosst et al., 2019). However, most of these are not formulated for sequence modeling. Here, we provide a general formulation of contrastive loss for sequences, to facilitate our subsequent discussion of GRPO and DPO.

**Definition 3.1** (Contrastive loss for sequences). Let $\pi_\theta$ be a probabilistic model, $\mathcal{D}$ an arbitrary data distribution, $\boldsymbol{x} \sim \mathcal{D}$ be the anchor sequence, and $\mathcal{D}^+(\cdot \mid \boldsymbol{x})$ and $\mathcal{D}^-(\cdot \mid \boldsymbol{x})$ be the positive and negative distributions conditioned on $\boldsymbol{x}$. We call $\boldsymbol{y}_i^+ \sim \mathcal{D}^+$ a positive sequence and $\boldsymbol{y}_j^- \sim \mathbb{D}^-$ a negative sequence w.r.t. $\boldsymbol{x}$. Let $\boldsymbol{y}_{i,t}$ denote the $t$-th token of $\boldsymbol{y}_i$, which is generated autoregressively. Given $N$ positive samples and $M$ negative samples, we say a differentiable loss function $\mathcal{L}$ is *contrastive* if its gradient has the following form:

$$\nabla_\theta \mathcal{L} =$$

$$- \mathbb{E}_{[\boldsymbol{x} \sim \mathcal{D}]} \left( \frac{1}{N} \sum_{i=1}^{N} \sum_{t=1}^{|\boldsymbol{y}_i^+|} c_{i,t}^+ \nabla_\theta \pi_\theta(\boldsymbol{y}_{i,t}^+|\boldsymbol{y}_{i,<t}^+, \boldsymbol{x}) - \frac{1}{M} \sum_{j=1}^{M} \sum_{t=1}^{|\boldsymbol{y}^-|} c_{j,t}^- \nabla_\theta \pi_\theta\left(\boldsymbol{y}_{j,t}^-|\boldsymbol{y}_{j,<t}^-, \boldsymbol{x}\right) \right) , \quad (7)$$

where $c_{i,t}^+$ and $c_{j,t}^-$ are token-level coefficients whose explicit forms depend on the specific design of the loss; they may be constants or functions conditioned on the inputs.

In the case of sequences, contrastive losses with sequence-level (i.e., example-level) coefficients can always be equivalently expressed using token-level coefficients. Thus, we adopt the definition with token-level coefficients. Eq. (7) represents the practical contrastive loss used in practice, which serves as an empirical estimate of the true theoretical gradient defined in Eq. (8).

$$\nabla_\theta \mathcal{L}^* =$$

$$- \mathbb{E}_{[\boldsymbol{x} \sim \mathcal{D}]} \left( \mathbb{E}_{[\boldsymbol{y}_i \sim \mathcal{D}^+]} \sum_{t=1}^{|\boldsymbol{y}_i^+|} c_{i,t}^+ \nabla_\theta \pi_\theta(\boldsymbol{y}_{i,t}^+|\boldsymbol{y}_{i,<t}^+, \boldsymbol{x}) - \mathbb{E}_{[\boldsymbol{y}_j \sim \mathcal{D}^-]} \sum_{t=1}^{|\boldsymbol{y}_j^-|} c_{j,t}^- \nabla_\theta \pi_\theta\left(\boldsymbol{y}_{j,t}^-|\boldsymbol{y}_{j,<t}^-, \boldsymbol{x}\right) \right) \quad (8)$$

A larger $N$ and $M$ in Eq. (7) can provide a more accurate estimate of the true gradient (Eq. (8)).

## 3.2 GRPO: N-VS-M CONTRASTIVE LEARNING

Revisiting GRPO (Eq. (5)), we can see that GRPO is performing dynamic $N$-vs-$M$ contrastive learning, where $G = N + M$ and the specific values of $N$ and $M$ are determined by the sampled trajectories. Let $G_q^+$ and $G_q^-$ denote the numbers of correct and incorrect trajectories of prompt $q$, respectively, in the $G$ sampled trajectories. $\hat{p}_{\theta_{\text{old}},q} = G_q^+/G$ is the proportion of correct trajectories in the sampled $G$ trajectories, which approximates the probability of correct $p_{\theta_{\text{old}},q}$ given the policy $\pi_{\theta_{\text{old}},q}$ on the prompt $q$. In the following discussion, we drop the subscript $\theta_{\text{old}}$ for simplicity.

We have the following equation for the GRPO objective function:

$$\mathcal{J}_{\text{GRPO}}(\theta, G) = \mathbb{E}_{[q \sim \mathcal{Q}; \{o_j^+, o_k^-\}_{j,k}^G \sim \pi_{\theta_{\text{old}}}(\cdot|q)]}$$

$$\sqrt{\widehat{\text{Var}}_G(q)} \left[ \frac{1}{G_q^+} \sum_{j=1}^{G_q^+} \frac{1}{|o_j^+|} \sum_{t=1}^{|o_j^+|} \mathcal{C}_\epsilon^+ \left( \frac{\pi_\theta(o_{j,t}^+|o_{j,<t}^+, q)}{\pi_{\theta_{\text{old}}}(o_{j,t}^+|o_{j,<t}^+, q)} \right) - \frac{1}{G_q^-} \sum_{k=1}^{G_q^-} \frac{1}{|o_k^-|} \sum_{t=1}^{|o_k^-|} \mathcal{C}_\epsilon^- \left( \frac{\pi_\theta(o_{k,t}^-|o_{k,<t}^-, q)}{\pi_{\theta_{\text{old}}}(o_{k,t}^-|o_{k,<t}^-, q)} \right) \right],$$

$$(9)$$

where $o_j^+$ and $o_k^-$ denotes the rollouts with true and false rewards, respectively; $\widehat{\text{Var}}_G(q) = (1 - \hat{p}_q)\hat{p}_q$ is the empirical standard deviation of a group of $G$ samples from Bernoulli$(p_q)$, which is the distribution of rewards in the verifiable setting. For simplicity, we use denote the upper clipping and lower clipping as $\mathcal{C}_\epsilon^+(x) = \min[x, 1 + \epsilon]$ and $\mathcal{C}_\epsilon^- = \max[x, 1 - \epsilon]$

**Proposition 3.2.** *Maximizing GRPO objective is equivalent to minimizing a $N$-vs-$M$ contrastive loss.*

*Proof of Proposition 3.2.* Eq. (9) has the following derivatives:

$$\nabla_\theta \mathcal{J}_{\text{GRPO}}$$

$$= \mathbb{E}_{[q \sim \mathcal{Q}]} \sqrt{\widehat{\text{Var}}_G(q)} \left( \frac{1}{G_q^+} \sum_{j=1}^{G_q^+} \sum_t^{|o_j^+|} \frac{\mathbb{1}_{j,t}^\epsilon \nabla_\theta \pi_\theta(o_{j,t}^+|o_{j,<t}^+, q)}{|o_j^+|\pi_{\theta_{\text{old}}}(o_{j,t}^+|o_{j,<t}^+, q)} - \frac{1}{G_q^-} \sum_{k=1}^{G_q^-} \sum_t^{o_k^-} \frac{\mathbb{1}_{k,t}^\epsilon \nabla_\theta \pi_\theta(o_{k,t}^-|o_{k,<t}^-, q)}{|o_k^-|\pi_{\theta_{\text{old}}}(o_{k,t}^-|o_{k,<t}^-, q)} \right)$$

$$= \mathbb{E}_{[q \sim \mathcal{Q}]}$$

$$\left[ \underbrace{\frac{1}{G_q^+} \sum_{j=1}^{G_q^+} \sum_t^{|o_j^+|} c(o_{j,t}^+|o_{j,<t}^+, q) \nabla_\theta \pi_\theta(o_{j,t}^+|o_{j,<t}^+, q)}_{\text{Positive}} - \underbrace{\frac{1}{G_q^-} \sum_{k=1}^{G_q^-} \sum_t^{|o_k^-|} c(o_{k,t}^-|o_{k,<t}^-, q) \nabla_\theta \pi_\theta(o_{k,t}^-|o_{k,<t}^-, q)}_{\text{Negative}} \right]$$

$$(10)$$

where $\mathbb{1}_{j,t}^\epsilon$ is an indicator function if the token $o_{j,t}$ clipped; $c(o_{i,t}|o_{i,<t}, q) := \frac{\sqrt{\widehat{\text{Var}}(q)}\mathbb{1}_{i,t}^\epsilon}{|o_i|\pi_{\theta_{\text{old}}}(o_{i,t}|o_{i,<t}, q)}$. Therefore, maximizing the GRPO objective is equivalent to minimizing a $N$-vs-$M$ contrastive loss (Def. 3.1), which completes the proof. $\square$

## 3.3 DPO: 1-VS-1 CONTRASTIVE LEARNING

**Proposition 3.3.** *The DPO loss is a $1$-vs-$1$ contrastive loss.*

*Proof of Proposition 3.3.* The DPO loss (Eq. (4)) has the following derivatives:

$$\nabla_\theta \mathcal{L}_{\text{DPO}} =$$

$$- \beta \mathbb{E}_{[(q,o^+,o^-) \sim \mathcal{D}_{\text{DPO}}]} \left[ \sigma(\hat{r}_\theta(q, o^-) - \hat{r}_\theta(q, o^+)) \left( \nabla_\theta \log \pi_\theta(o^+|q) - \nabla_\theta \log \pi_\theta(o^-|q) \right) \right]$$

$$= - \mathbb{E}_{[(q,o^+,o^-) \sim \mathcal{D}_{\text{DPO}}]} \left[ \underbrace{\sum_t^{|o^t|} c(o_t^+|o_{<t}^+, q) \nabla_\theta \pi_\theta(o_t^+|o_{<t}^+, q)}_{\text{Positive}} - \underbrace{\sum_t^{|o^-|} c(o_t^+|o_{<t}^+, q) \nabla_\theta \pi_\theta(o_t^-|o_{<t}^-, q)}_{\text{Negative}} \right]$$

$$(11)$$

where $\hat{r}_\theta = \beta(x, y) \log \frac{\pi_\theta(y|x)}{\pi_{\text{ref}}(y|x)}$; and $c(o_t|o_{<t}, q) := \frac{\beta\sigma(\hat{r}_\theta(q, o^-) - \hat{r}_\theta(q, o^+))}{\pi_\theta(o|q)}$, aligning with Def. 3.1.

$\square$

## 3.4 GRPO V.S. DPO

Viewed through the lens of contrastive learning, both GRPO and DPO implement direct preference optimization: they promote preferred trajectories while discouraging dispreferred ones. In DPO, preferences are specified by human annotations, while in GRPO, they are determined by the reward model.

The primary differences between the GRPO and DPO objectives are as follows: (1) they adopt different designs of token coefficient $c$; (2) GRPO incorporates multiple positive and negative examples in a dynamic $N$-vs-$M$ setting, whereas DPO utilizes exactly one positive and one negative example per case.

The first distinction arises from the training paradigms. DPO operates within an offline RL framework using pre-collected, human-annotated data. In contrast, GRPO functions in an online RL context where data is sampled from the policy network, employing importance sampling with clipping to stabilize off-policy optimization. To apply DPO in an online RL setting, similar techniques for ensuring learning stability and effectiveness are expected to be introduced.

The second key difference lies on the number of positive and negative samples used to estimate the true gradient of the contrastive loss in Eq. (8). The number of positive and negative examples used in contrastive learning mainly impacts the variance of the gradient estimator. Nevertheless, the gradient remains an unbiased estimate of the true objective even with only a single positive and negative sample, allowing effective optimization to proceed.

Given that the majority of computation in LLM post-training is spent on rollout generation – accounting for up to 70% of total training time (Liu et al., 2025) – it is natural to ask whether a DPO-like GRPO—using only one positive and one negative example—can still be effective.

## 4 RETHINKING GROUP SIZE: ARE TWO ENOUGH?

From the standpoint of reward normalization, using a group size of two appears suboptimal, as it is viewed as an invalid and unreliable estimate of statistics. However, from the contrastive learning perspective, GRPO with a group size of two – denoted as 2-GRPO – functions similarly to standard GRPO, as the choice of group size primarily affects the variance of the gradient estimate, rather than the fundamental learning dynamics.

Before moving on to our empirical evaluation, we present a more thorough theoretical justification for the design choices underlying 2-GRPO in this section. 2-GRPO, i.e., GRPO with group size $G = 2$, is formally defined as:

$$\mathcal{J}_{\text{GRPO}}(\theta, G) = \mathbb{E}_{[q\sim\mathcal{Q};(o_1,o_2)\sim\pi_{\theta_{\text{old}}}(\cdot|q)]}$$

$$\mathbb{1}(r_1 \neq r_2) \cdot \frac{1}{2} \left[ \frac{1}{|o_1|} \sum_{t=1}^{|o_1|} \mathcal{C}_\epsilon^+ \left( \frac{\pi_\theta(o_{1,t}|o_{1,<t}, q)}{\pi_{\theta_{\text{old}}}(o_{1,t}|o_{1,<t}, q)} \right) - \frac{1}{|o_2|} \sum_{t=1}^{|o_2|} \mathcal{C}_\epsilon^- \left( \frac{\pi_\theta(o_{2,t}|o_{2,<t}, q)}{\pi_{\theta_{\text{old}}}(o_{2,t}|o_{2,<t}, q)} \right) \right],$$

$$(12)$$

For the simplicity of notation, without loss of generality, we assume $o_1$ to be the positive sample and $o_2$ to be the negative sample when their rewards are not the same. In other words, the advantage estimation of 2-GRPO is a flawed normalization: $A^+ = 1, A^- = -1$ for a positive-negative pair and $A^+ = A^- = 0$ otherwise.

Advantage estimation serves two primary functions in RL optimization: **credit assignment** and **gradient variance reduction**. We will discuss them in the following sections.

### 4.1 ADVANTAGE SHAPING IN STOCHASTIC OPTIMIZATION

In standard GRPO, advantage estimation assigns credit to positive and negative samples based on the empirical success rate $\hat{p}_q$, which estimates the true correctness probability $p_q$ (the larger group-size,

the better estimation). In 2-GRPO, this mechanism appears spurious given the group size of two. However, in this section, we demonstrate that under stochastic optimization, advantage estimation in 2-GRPO still preserves proper credit assignment.

**Proposition 4.1.** *Given a constant $p \in (0, 1)$ and a small positive constant $\epsilon$, we consider two scenarios below:*

- **Case 1**: *Consider $X_1, \cdots, X_{2N} \overset{i.i.d.}{\sim} Bernoulli(p)$. Let $Y_i = \frac{X_i - \hat{\mu}}{\hat{\sigma} + \epsilon}$, where $\hat{\mu} = \frac{1}{2N} \sum_{i=1}^{2N} X_i$ and $\hat{\sigma} = \sqrt{\frac{1}{2N} \sum_{i=1}^{2N} (X_i - \hat{\mu})^2}$. Then, it follows that*

$$\lim_{\epsilon \to 0} \lim_{N \to \infty} \mathbb{E}[Y_i | X_i = x] = \frac{x - p}{\sqrt{p(1-p)}}. \tag{13}$$

- **Case 2**: *Consider $N$ pairs of $(X_{i,1}, X_{i,2})$ with each $X_{i,j} \overset{i.i.d.}{\sim} Bernoulli(p)$. Let $Y_{i,j} = \frac{X_{i,j} - \hat{\mu}_i}{\hat{\sigma}_i + \epsilon}$, where $\hat{\mu}_i = \frac{1}{2}(X_{i,1} + X_{i,2})$ and $\hat{\sigma}_i = \sqrt{\frac{1}{2} \sum_{j=1}^{2} (X_{i,j} - \hat{\mu}_i)^2}$. Then, it follows that*

$$\lim_{\epsilon \to 0} \lim_{N \to \infty} \mathbb{E}[Y_{i,j} | X_{i,j} = x] = x - p. \tag{14}$$

*The $\lim_{\epsilon \to 0} \lim_{N \to \infty} \mathbb{E}[Y_{i,j} | X_{i,j} = x]$ differs from $\lim_{\epsilon \to 0} \lim_{N \to \infty} \mathbb{E}[Y_i | X_i = x]$ by a scaling factor $\frac{1}{\sqrt{p(1-p)}}$.*

In Proposition 4.1 (proof in Appendix A.3), **Case 1** corresponds to regular GRPO with sufficiently large group size; in this case, $\mathbb{E}[Y_i | X_i = 1]$ and $\mathbb{E}[Y_i | X_i = 0]$ are, respectively, the advantage estimates of positive and negative trajectories given a prompt, dependent on the success probability $p_q$. A large $G$ will lead to a better estimate of the success probability $p_q$. **Case 2** corresponds to 2-GRPO, where $\mathbb{E}[Y_{i,j} | X_{i,j} = 1]$ and $\mathbb{E}[Y_{i,j} | X_{i,j} = 0]$ are advantage estimates, which are also dependent on the success rate $p_q$, amortizing over multiple stochastic updates.

The advantage estimates in 2-GRPO differ from those in standard GRPO solely by a scaling factor. It is worth mentioning that whether this scaling is beneficial remains an open question (Li et al., 2025).

### 4.2 Variance of Gradient Estimate

As mentioned in Sec. 3.1, reducing the group size (i.e., the number of positive and negative examples) will lead to higher variance of the gradient estimate for each prompt. However, this conclusion ignores the fact that we are doing mini-batch optimization.

In this section, we discuss the practical gradient variance under mini-batch optimization. To clarify this, we focus on the optimization phase of RL and denote the sampled rollouts as training data for notation simplicity.

Firstly, we can provide a definition of gradient variance, followed by a lemma for empirical gradient estimation.

**Definition 4.2** (Gradient Variance). Without loss of generality, let $\{\boldsymbol{x}_i\}_{i=1}^B$ be a batch of $B$ random variables (r.v.'s), where each $\boldsymbol{x}_i$ are i.i.d. $\boldsymbol{x} \sim \mathcal{D}$, and let $g(\boldsymbol{x}_i) = \nabla_\theta L_\theta(\boldsymbol{x}_i)$ denote the gradient of $L_\theta(\boldsymbol{x}_i)$ w.r.t. $\theta$. Define the empirical batch gradient $\hat{g}_B = \frac{1}{B} \sum_{i=1}^B g(\boldsymbol{x}_i)$. Note that $g(\boldsymbol{x}_i)$ and $\hat{g}_B$ are dependent r.v.'s of $\boldsymbol{x}_i$ and $\{\boldsymbol{x}_i\}_{i=1}^B$, respectively. We denote the expectation of gradient $\bar{g} = \mathbb{E}_{\boldsymbol{x} \sim \mathcal{D}}[g(\boldsymbol{x})]$. The variance of the gradient estimate over the batch is then defined as:

$$\text{Var}(\hat{g}_B) = \text{Var}_{\{\boldsymbol{x}_i\}_i^B}(\hat{g}_B) = \mathbb{E}_{\{\boldsymbol{x}_i\}_i^B}\left((\hat{g}_B - \bar{g})^2\right). \tag{15}$$

**Lemma 4.3.** *Let $\{\boldsymbol{x}_i\}_{i=1}^{B_1}, \{\boldsymbol{x}_i\}_{i=1}^{B_2}$ be two batches of $B_1$ and $B_2$ r.v.'s, respectively. Let $\hat{g}_{B_1}, \hat{g}_{B_2}$ denote the empirical batch gradients of these two batches, respectively. If $B_1 < B_2$, then $\text{Var}[\hat{g}_{B_1}] > \text{Var}[\hat{g}_{B_2}]$.*

*Proof of Lemma 4.3.*

$$\text{Var}(\hat{g}_B) = \text{Var}_{\{\boldsymbol{x}_i\}_i^B}\left(\frac{1}{B} \sum_i^B g(\boldsymbol{x}_i)\right) = \frac{1}{B^2}\left(\sum_i^B \text{Var}_{\boldsymbol{x}_i}(g(\boldsymbol{x}_i))\right) = \frac{\text{Var}_{\boldsymbol{x}}(g(\boldsymbol{x}))}{B}, \tag{16}$$

where the second and third equalities are obtained by the properties of independence and identity in i.i.d. data, respectively. By the above equation, increasing $B$ decreases Var. $\qquad\square$

At first glance, decreasing the group size in Eq. (10) seems to increase the variance of the gradient for each prompt. However, we have omitted the fact that the actual gradient calculation is obtained across different prompts in a mini-batch fashion.

In Lemma 4.2, we show that the larger batch size $B$ will naturally lead to the lower variance of the gradient. Note that $B$ is the **number of rollouts** rather than the **number of prompts** in a mini-batch.

The actual calculation of GRPO is:

$$\widehat{\mathcal{J}}_{\text{GRPO}}(\theta, G, Q) = \frac{1}{QG} \sum_{j=1}^{Q} \sum_{i=1}^{G} A_{ij} \pi_{\theta}^{\text{GRPO}}(o_{ij}|q_j),\tag{17}$$

where $\pi_{\theta}^{\text{GRPO}}(o^+|q) = \frac{1}{G} \sum_{i=1}^{G} \frac{1}{|o_i|} \sum_{t=1}^{|o_i|} \mathcal{C}_\epsilon \left( A_{i,t} \frac{\pi_{\theta}(o_{i,t}|o_{i,<t},q)}{\pi_{\theta_{\text{old}}}(o_{i,t}|o_{i,<t},q)} \right)$ and $Q$ is the number of prompts in the mini-batch, and the batch size w.r.t the number of rollouts is $B = QG$. When we decrease $G$, we can increase $Q$ to compensate to retain the same $B$. It is worth noting that, since the total number of prompts in the dataset is fixed, increasing $Q$ does not increase the computational cost per training epoch.

While we do not claim that lower variance inherently guarantees better training outcomes, we demonstrate that 2-GRPO can achieve gradient variance comparable to regular GRPO by adjusting the number of prompts in each mini-batch.

## 4.3 EXPLORATION ON HARD QUESTIONS

Another common concern with using a small group size in GRPO (e.g., $G = 2$) is that it may perform poorly on difficult prompts, where multiple attempts are often needed to produce a correct answer. The intuition behind this is that a smaller group provides fewer opportunities to sample a correct response within a single batch, potentially slowing down learning.

Under a fixed computational budget – 2-GRPO and 16-GRPO explore approximately the same total number of rollouts across all training epochs – the overall probability of sampling a correct answer under 2-GRPO is not lower than 16-GRPO, according to the Proposition 4.4.

**Proposition 4.4.** *Let $p_i \in [0, 1]$ denote the probability that a single rollout under the policy $\pi_i$ produces a correct answer. Then:*

1. *The probability of obtaining at least one correct answer in $2m$ independent rollouts with policy $\pi_0$ is*

$$P_{2m} = 1 - (1 - p_0)^{2m}.\tag{18}$$

2. *The probability of obtaining at least one correct answer when performing $m$ consecutive trials of $2$ independent rollouts each, with the corresponding policy $[\pi_0, \pi_1, \cdots, \pi_{m-1}]$ is*

$$P_{m\times 2} = 1 - \prod_{i=0,\cdots m-1} (1 - p_i)^2 \geq 1 - (1 - p_0)^{2m} = P_{2m}\tag{19}$$

   *when we have $p_i \geq p_0, \forall i > 0$.*

*Note that the assumption $p_i \geq p_0, \forall i > 0$ is prevailing, as we assume that the reasoning ability of LLM can be improved by RL post-training.*

Proposition 4.4 indicates that for hard questions, 2-GRPO will not breakdown compared to 16-GRPO, given the same total rollouts traversed. It is worth mentioning that, due to its greater number of policy updates, 2-GRPO may have a higher probability of getting a correct output for a difficult question and is more adaptive to capture more nuanced update requirements for different questions.

## 5 EXPERIMENTS

### 5.1 EXPERIMENT DETAILS

**Tasks and Training Framework**    Following prior studies, we consider mathematical tasks as representative instances of RLVR to verify our hypothesis, given their demonstrated transferability to a broad range of other tasks (Yu et al., 2025). For training, we adopt the *verl* framework (Sheng et al., 2025) and utilize the built-in implementation of GRPO (Shao et al., 2024) as the baseline algorithm.

**Dataset and Baselines**    Following prior work (Chu et al., 2025), we employ Qwen-2.5-Math-1.5B (Qwen-1.5B) and Qwen-2.5-Math-7B (Qwen-7B) (Yang et al., 2025) as base models. Both models are post-trained via RL on the MATH (Hendrycks et al., 2021) and DAPO-Math-17k (Yu et al., 2025) datasets, and evaluated on MATH-500 (Hendrycks et al., 2021), AMC23, Minerva Math (Lewkowycz et al., 2022), AIME-2025, and OlympiadBench (He et al., 2024). For DAPO-Math-17k dataset, we randomly sample 7.5k questions from the original data to form a subset for training in order to align with the size of MATH. In addition, we assess the proposed method on DeepSeek-R1-Distill-Qwen-1.5B (DS-1.5B) (DeepSeek-AI, 2025), which is post-trained on MATH. Owing to computational constraints, we do not extend its post-training to DAPO-Math-17k. All 1.5B models are trained on 4 GPUs. Qwen-7B is trained on 8 GPUs. We evaluate model performance using two metrics: Mean@32, the average accuracy across 32 i.i.d. samples, and Pass@32, which measures whether a problem is solved in at least one of those 32 attempts.

**Hyper-parameters**    We mainly follow the default configuration of the *verl* framework. For sampling parameters in training generation, we set temperature to 1, top-p to 1 to encourage exploration, sequence length to 4096 for Qwen-series model and 8192 for DS-1.5B. For sampling parameters in test generation, we set temperature to 0.7, top-p to 0.8, top-k to 20 and sequence length to 4096 for all models. For optimization, training employs the Adam optimizer (Kingma, 2014) with a constant learning rate and a linear warm-up over the first 10 steps. For GRPO hyper-parameters, we set the clip ratio high to $0.28$ and clip ratio lower to $0.2$ following DAPO (Yu et al., 2025). All models are trained for 10 epochs. The baseline method, 16-GRPO, is trained with batch sizes of 32 (32 prompts and 16 rollouts per prompt) and a learning rate $1 \times 10^{-6}$. As discussed in Sec. 4.2, we trained 2-GRPO with a larger batch size of 256 (256 prompts and 2 rollouts per prompt). Both case will have 512 rollouts in each mini-batch of training. Since we have fewer update steps due to the larger batch size, we adjust the learning rate of 2-GRPO to $8 \times 10^{-6}$ based on the linear relationship of learning rate and batch size (Goyal et al., 2017).

**Goal of Experiment**    Building on the theoretical justification for 2-GRPO, we seek to empirically assess its validity in RLVR. We anticipated that *2-GRPO will exhibit better efficiency*—with respect to computational resources and/or wall-clock time—while maintaining the same performance as regular GRPO (16-GRPO).

### 5.2 MAIN EXPERIMENTS

As shown in Table 1, 2-GRPO requires at least $70\%$ less wall-clock time than 16-GRPO while achieving comparable performance. The models are post-trained on the MATH and DAPO-Math-Sub datasets and evaluated on five widely-used mathematical reasoning benchmarks, representing an out-of-distribution evaluation. This setting imposes stringent requirements on the generalization ability of the post-trained models. Notably, 2-GRPO is optimized with only $0.15$ million generated rollouts — just $12.5\%$ of the $1.2$ million rollouts utilized by 16-GRPO. [2] These results provide strong corroboration of our theoretical finding that reducing group size preserves performance while substantially improving efficiency. To further support this statement, we conduct ablation study on various k-GRPO ($k = 4, 8$) in Appendix B.2.

### 5.3 VISUALIZATION

In Sec. 5.2, we present empirical results comparing 2-GRPO and 16-GRPO. However, the out-of-distribution evaluation setting may not fully reflect the post-training with 2-GRPO, as the distri-

---

[2]Appendix B.1 discusses the relationship between the total number of rollouts and computational cost.

Table 1: 2-GRPO v.s. 16-GRPO: post-trained on MATH/DAPO-Math-Sub and evaluated on five mathematical reasoning benchmarks. M/P@32 stands for Mean@32 and Pass@32. $G$ is the group size. $\Delta$ denotes the difference $16 \rightarrow 2$.

| M/P@32 ↑ | $G$ | Time (h) ↓ | MATH-500 | AMC 2023 | Minerva Math | AIME 2025 | Olympiad Bench |
|---|---|---|---|---|---|---|---|
| | | | *Post-training on MATH dataset* | | | | |
| Qwen-1.5B | w/o | - | 31.83 / 81.92 | 34.30 / 79.23 | 5.33 / 28.91 | 3.64 / 22.31 | 15.40 / 37.16 |
| | 2 | 2.05 | 69.28 / 87.43 | 49.53 / 81.76 | 16.25 / 33.26 | 9.48 / 32.88 | 22.31 / 37.24 |
| | 16 | 8.53 | 70.24 / 87.24 | 51.25 / 83.46 | 16.84 / 33.46 | 10.10 / 35.82 | 23.11 / 37.82 |
| | $\Delta$ | -75.96% | -0.96 / +0.19 | -1.71 / -1.70 | -0.59 / -0.19 | -0.62 / -2.94 | -0.80 / -0.58 |
| Qwen-7B | w/o | - | 47.16 / 85.95 | 38.36 / 85.29 | 5.99 / 31.10 | 5.00 / 25.17 | 9.83 / 34.30 |
| | 2 | 2.43 | 75.23 / 89.77 | 64.60 / 81.53 | 23.13 / 38.45 | 12.81 / 38.85 | 26.39 / 40.20 |
| | 16 | 9.30 | 75.90 / 88.24 | 61.79 / 80.77 | 22.81 / 37.68 | 13.23 / 34.22 | 25.99 / 40.11 |
| | $\Delta$ | -73.87% | -0.67 / +1.53 | +2.81 / +0.76 | +0.32 / +0.77 | -0.42 / +4.63 | +0.40 / 0.09 |
| DS-1.5B | w/o | - | 65.11 / 84.90 | 44.14 / 73.86 | 14.64 / 32.80 | 22.40 / 42.79 | 20.07 / 33.23 |
| | 2 | 7.07 | 74.36 / 88.85 | 56.95 / 88.63 | 21.28 / 38.34 | 24.89 / 46.79 | 33.69 / 45.86 |
| | 16 | 38.40 | 75.98 / 89.16 | 58.91 / 87.26 | 21.76 / 38.29 | 26.97 / 56.36 | 35.39 / 47.05 |
| | $\Delta$ | -81.6% | -1.62 / -0.31 | -1.96 / +1.38 | -0.48 / -0.05 | -2.08 / -9.56 | -1.70 / -1.19 |
| | | | *Post-training on DAPO-Math-Sub dataset* | | | | |
| Qwen-1.5B | w/o | - | 31.83 / 81.92 | 34.30 / 79.23 | 5.33 / 28.91 | 3.64 / 22.31 | 15.40 / 37.16 |
| | 2 | 2.12 | 68.81 / 87.36 | 52.19 / 85.77 | 16.79 / 33/61 | 8.13 / 29.33 | 23.52 / 39.29 |
| | 16 | 13.30 | 70.66 / 87.04 | 56.56 / 85.54 | 18.00 / 34.16 | 9.58 / 32.31 | 24.56 / 39.19 |
| | $\Delta$ | -84.06% | -1.85 / +0.32 | -4.37 / +0.23 | -1.21 / +0.71 | -2.50 / -2.98 | -1.04 / +0.10 |
| Qwen-7B | w/o | - | 47.16 / 85.95 | 38.36 / 85.29 | 5.99 / 31.10 | 5.00 / 25.17 | 9.83 / 34.30 |
| | 2 | 3.63 | 77.43 / 90.51 | 64.84 / 91.59 | 21.95 / 38.05 | 14.58 / 33.03 | 29.86 / 45.24 |
| | 16 | 17.68 | 77.35 / 88.79 | 69.69 / 87.31 | 24.45 / 40.04 | 14.27 / 33.73 | 28.86 / 39.84 |
| | $\Delta$ | -79.47% | +0.08 / +1.72 | -4.85 / +4.28 | -2.50 / -1.99 | +0.31 / -0.70 | +1.00 / +5.4 |

bution shift could obscure the underlying performance differences. Therefore, in this section, we visualize the reward and evaluation scores on the MATH dataset to demonstrate the in-distribution generalization of the post-trained models using 2-GRPO in comparison to 16-GRPO. [3]

The figures presented in Fig. 1 and Fig. 2 illustrate the performance of Qwen-2.5-Math-1.5B and Qwen-2.5-Math-7B, respectively. As depicted, the reward and evaluation scores for 2-GRPO are comparable to those of 16-GRPO, indicating that the in-distribution generalization of the post-trained models using 2-GRPO is on par with that of 16-GRPO.

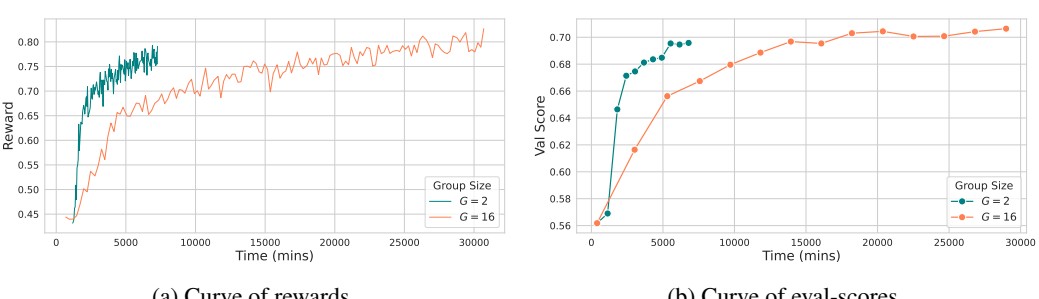

| (a) Curve of rewards. | (b) Curve of eval-scores. |
|---|---|

Figure 1: Qwen-1.5B: Visualization of reward and evaluation scores on the MATH dataset.

## 6 DISCUSSION

**Stronger Efficiency** There remains potential for further enhancements of 2-GRPO in efficiency. In 2-GRPO, many rollouts generated are ultimately assigned zero advantage, which actually do not

---

[3]The DAPO dataset does not provide a test set.

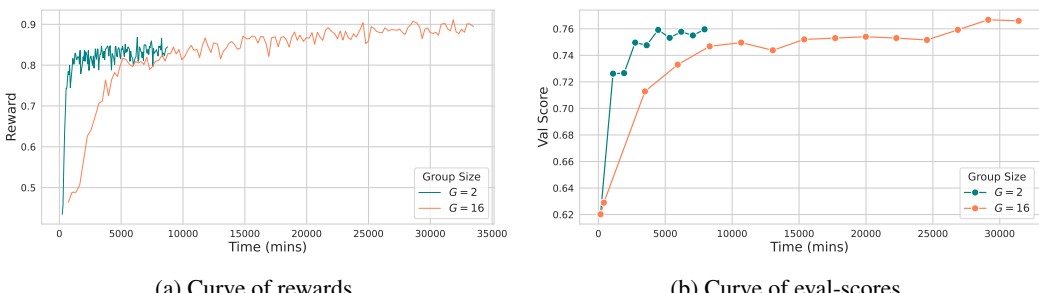

(a) Curve of rewards.                    (b) Curve of eval-scores.

Figure 2: Qwen-7B: Visualization of reward and evaluation scores on the MATH dataset.

demand the computation of gradients. Consequently, a more advanced implementation could optimize these computations during the training phase. It is important to note that, as discussed in Sec. 4.1, these zero-advantage rollouts are still necessary for accurate advantage estimation. Therefore, we must simulate the contributions of these zero-advantage rollouts during the training phase rather than simply discarding them after the inference phase.

**2-GRPO is a Quantization of GRPO**   An alternative perspective on 2-GRPO is that it serves as a quantization of standard GRPO, wherein the candidate values for advantages are discretized to $-1, 0, 1$. Nevertheless, due to the stochastic nature of neural network optimization, 2-GRPO is capable of approximating continuous advantage values effectively, provided that a sufficiently large number of training steps are employed.

**Data Efficiency**   The quantized nature of 2-GRPO inherently results in the rejection of a number of generated rollouts. While this characteristic enhances computational efficiency, it may concurrently compromise data efficiency – numerous rollouts are discarded when the policy exhibits either exceptionally strong performance. This limitation in data efficiency could impede the ability of the policy post-trained by 2-GRPO to attain near-optimal performance. This observation might be mitigated by replaying buffered negative samples, where we leave this direction to future exploration.

**Conclusion**   In this work, we present a theoretical analysis of GRPO from a contrastive learning perspective, establishing a key conceptual connection between GRPO and DPO. This perspective offers a new lens for understanding GRPO: we argue that for group-relative advantage estimation, strict normalization is not essential; rather, the contrastive learning structure is the fundamental driver. Building on this insight, we propose 2-GRPO, a DPO-inspired variant with a group size of two that achieves significant efficiency gains while maintaining comparable performance.

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

APPENDIX

# A THEOREMS

## A.1 REVEAL GRPO AS CONTRASTIVE

*Details of Sec. 3.* In the RLVR setting, rewards are binary, which leads to binary advantages given a prompt. Let $A_q^+$, $A_q^-$ denote the positive and negative advantage, respectively. From Eq. (6), we can have

$$
\begin{aligned}
A_q^+ &= \frac{1 - \hat{p}_q}{\sqrt{\hat{p}_q(1 - \hat{p}_q)}} = \sqrt{\frac{1 - \hat{p}_q}{\hat{p}_q}} \; , \\
A_q^- &= \frac{0 - \hat{p}_q}{\sqrt{\hat{p}_q(1 - \hat{p}_q)}} = -\sqrt{\frac{\hat{p}_q}{1 - \hat{p}_q}} \; .
\end{aligned}
\tag{20}
$$

In Eq. (5), the clipping function is

$$
\text{clip}(x, 1 - \epsilon, 1 + \epsilon) = \begin{cases} x, & |x - 1| \leq \epsilon \\ 1 - \epsilon, & x < 1 - \epsilon \\ 1 + \epsilon, & x > 1 + \epsilon \end{cases} ,
\tag{21}
$$

which means that $x$ will be assigned to $1 - \epsilon$ $(1 + \epsilon)$ if $x$ is less (greater) than $1 - \epsilon$ $(1 + \epsilon)$. For simplifying notation, let $\mathcal{C}_\epsilon(x)$ denote $\min[x, \text{clip}(x, 1 - \epsilon, 1 + \epsilon)]$.

The key derivation is as follows:

$$
\mathcal{J}_{\text{GRPO}}(\theta)
$$

$$
= \mathbb{E}_{\substack{q \sim \mathcal{Q} \\ \{o_i\}_{i=1}^G \sim \pi_{\theta_{\text{old}}}(\cdot|q)}} \frac{1}{G} \sum_{i=1}^{G} \frac{1}{|o_i|} \sum_{t=1}^{|o_i|} \mathcal{C}_\epsilon \left( \frac{\pi_\theta(o_{i,t}|o_{i,<t}, q)}{\pi_{\theta_{\text{old}}}(o_{i,t}|o_{i,<t}, q)} A_{i,t} \right) ,
$$

$$
= \mathbb{E}_{\substack{q \sim \mathcal{Q} \\ \{o_j\}_{j=1}^{G^+} \sim \pi_{\theta_{\text{old}}}^+(\cdot|q) \\ \{o_k\}_{k=1}^{G^-} \sim \pi_{\theta_{\text{old}}}^-(\cdot|q)}}
$$

$$
\frac{1}{G} \left( \sum_{j=1}^{G^+} \frac{1}{|o_j|} \sum_{t=1}^{|o_j|} A_j^+ \mathcal{C}_\epsilon^+ \left( \frac{\pi_\theta(o_{j,t}|o_{j,<t}, q)}{\pi_{\theta_{\text{old}}}(o_{j,t}|o_{j,<t}, q)} \right) + \sum_{k=1}^{G^-} \frac{1}{|o_k|} \sum_{t=1}^{|o_k|} A_k^- \mathcal{C}_\epsilon^- \left( \frac{\pi_\theta(o_{k,t}|o_{k,<t}, q)}{\pi_{\theta_{\text{old}}}(o_{k,t}|o_{k,<t}, q)} \right) \right) ,
$$

$$
= \mathbb{E}_{\substack{q \sim \mathcal{Q} \\ \{o_j\}_{j=1}^{G^+} \sim \pi_{\theta_{\text{old}}}^+(\cdot|q) \\ \{o_k\}_{k=1}^{G^-} \sim \pi_{\theta_{\text{old}}}^-(\cdot|q)}}
$$

$$
A_q^+ \frac{G^+}{G} \frac{1}{G^+} \sum_{j=1}^{G^+} \frac{1}{|o_j|} \sum_{t=1}^{|o_j|} \mathcal{C}_\epsilon^+ \left( \frac{\pi_\theta(o_{j,t}|o_{j,<t}, q)}{\pi_{\theta_{\text{old}}}(o_{j,t}|o_{j,<t}, q)} \right) + A_q^- \frac{G^-}{G} \frac{1}{G^-} \sum_{k=1}^{G^-} \frac{1}{|o_k|} \sum_{t=1}^{|o_k|} \mathcal{C}_\epsilon^- \left( \frac{\pi_\theta(o_{k,t}|o_{k,<t}, q)}{\pi_{\theta_{\text{old}}}(o_{k,t}|o_{k,<t}, q)} \right) ,
$$

$$
= \mathbb{E}_{\substack{q \sim \mathcal{Q} \\ \{o_j\}_{j=1}^{G^+} \sim \pi_{\theta_{\text{old}}}^+(\cdot|q) \\ \{o_k\}_{k=1}^{G^-} \sim \pi_{\theta_{\text{old}}}^-(\cdot|q)}}
$$

$$
\sqrt{\widehat{\text{Var}}_G(q)} \left( \frac{1}{G^+} \sum_{j=1}^{G^+} \frac{1}{|o_j|} \sum_{t=1}^{|o_j|} \mathcal{C}_\epsilon^+ \left( \frac{\pi_\theta(o_{j,t}|o_{j,<t}, q)}{\pi_{\theta_{\text{old}}}(o_{j,t}|o_{j,<t}, q)} \right) - \frac{1}{G^-} \sum_{k=1}^{G^-} \frac{1}{|o_k|} \sum_{t=1}^{|o_k|} \mathcal{C}_\epsilon^- \left( \frac{\pi_\theta(o_{k,t}|o_{k,<t}, q)}{\pi_{\theta_{\text{old}}}(o_{k,t}|o_{k,<t}, q)} \right) \right) .
\tag{22}
$$

The second equation is obtained by dividing the trajectories into two groups: positive and negative. The third equation is obtained by the fact that all positive advantages are the same and that all negative advantages are the same. Since $A^+ \frac{G^+}{G} = \sqrt{\frac{1-\hat{p}}{\hat{p}}}\hat{p} = \sqrt{(1 - \hat{p})\hat{p}}$ and $A^- \frac{G^-}{G} = -\sqrt{(1 - \hat{p})\hat{p}}$,

we obtain Eq. (9). When $G \to \infty$, we have the following facts:

$$
\begin{aligned}
& \lim_{G \to \infty} G^+ = \infty \ , \\
& \lim_{G \to \infty} G^- = \infty \\
& \lim_{G \to \infty} \sqrt{(1 - \hat{p})\hat{p}} = \sqrt{(1 - p)p} \ , \\
& \lim_{G^+ \to \infty} \frac{1}{G^+} \sum_{j=1}^{G^+} f(o_j) = \mathbb{E}_{o_j \sim O_\theta^+} f(o_j) \ , \\
& \lim_{G^- \to \infty} \frac{1}{G^-} \sum_{k=1}^{G^-} f(o_k) = \mathbb{E}_{o_k \sim O_\theta^-} f(o_k) \ .
\end{aligned}
\tag{23}
$$

$\square$

### A.2 JUSTIFICATION OF PROPOSITION 3.2

Most of autoregressive LLMs adopt causal probability modelling that $\sum_t \log \pi_\theta(o_t|o_{<t}, q) = \log \pi_\theta(o|q)$. Then we have the following equation to describe the gradient of trajectory probability and the gradient of token probabilities:

$$
\nabla_\theta \pi_\theta(o|q) = \pi_\theta(o|q) \sum_t \frac{1}{\pi_\theta(o_t|o_{<t}, q)} \pi_\theta(o_t|o_{<t}, q)
$$

However, the original GRPO objective does not hold this property. Or one can consider GRPO using the mean-field assumption for probability modelling. Some papers believe that GRPO should be corrected by sequence level importance sampling (Zheng et al., 2025; Zhao et al., 2025; Pang & Jin, 2025). It is still an open question for the choice of important sampling for GRPO.

### A.3 PROOF OF PROPOSITION 4.1

*Proof.* **Case 1.** Notice that $\hat{\sigma} = \sqrt{\frac{1}{2N} \sum_{k=1}^{2N} (X_k - \hat{\mu})^2} = \sqrt{\hat{\mu}(1 - \hat{\mu})}$ and $\hat{\mu} = \frac{1}{2N} \sum_{k=1}^{2N} X_k$. Fix an index $i$ and condition on the event $\{X_i = x\}$ with $x \in \{0, 1\}$. In this case, by the strong law of large numbers and the continuous mapping theorem, we have $\hat{\mu} \overset{a.s.}{\to} p$ and $\hat{\sigma} \overset{a.s.}{\to} \sqrt{p(1 - p)}$. Thus, it follows that

$$
\lim_{\epsilon \to 0} \lim_{N \to \infty} \mathbb{E}[Y_i \mid X_i = x] = \frac{x - p}{\sqrt{p(1 - p)}}.
$$

**Case 2.** When $X_{i,1} = X_{i,2}$, we have $X_{i,j} = \hat{\mu}_i$ and $Y_{i,j} = 0$ for any $j \in \{1, 2\}$. When $X_{i,1} \neq X_{i,2}$, we have $\hat{\mu}_i = 0.5$, $\hat{\sigma}_i = 0.5$, and $Y_{i,j} = \frac{2X_{i,j} - 1}{1 + 2\epsilon}$. By the law of total expectation, it follows that

$$
\mathbb{E}[Y_{i,j} \mid X_{i,j} = 1] = \frac{1 - p}{1 + 2\epsilon}, \qquad \mathbb{E}[Y_{i,j} \mid X_{i,j} = 0] = \frac{-p}{1 + 2\epsilon}.
$$

Thus, we have

$$
\lim_{\epsilon \to 0} \mathbb{E}[Y_{i,j} \mid X_{i,j} = x] = x - p.
$$

$\square$

## B EXPERIMENTS

### B.1 THE CONNECTION BETWEEN TRAINING ROLLOUTS AND COMPUTATIONAL COST

In Sec. 5.2, the total number of rollouts generated and utilized during training is adopted as a metric for comparing the computational cost of different methods.

The rationale for this choice is as follows. A principled measure of computational cost in the context of RL post-training is the number of floating-point operations (FLOPs) performed. Unlike wall-clock time, which is susceptible to variations arising from software implementation details (e.g.,

optimization of training libraries) and hardware characteristics (e.g., GPU/CPU architecture, I/O throughput), FLOPs provide a more direct and stable measure of computational effort.

For a fixed base model and the same type of RL algorithm (GRPO in our case), the FLOPs required for a single forward or backward pass with one input prompt can be considered constant, for both the generation and training phases. Accordingly, the total number of rollouts executed during training is directly proportional to the FLOPs executed, thereby serving as a theoretically justified and consistent proxy for computational cost.

## B.2 Ablation Study on the Group Size

We conducted an ablation study on the effect of group size. In this experiment, the batch size was fixed at 32 and the learning rate at $1 \times 10^{-6}$, following the configuration of the standard GRPO (16-GRPO). [4] Only the group size was varied in order to isolate and evaluate its impact.

Table 2: Ablation study on group size $G$: post-trained on MATH and DAPO, respectively, and evaluated on five mathematical reasoning benchmarks. M/P@32 stands for Mean@32 and Pass@32.

| M/P@32 ↑ | $G$ | Time (h) ↓ | MATH-500 | AMC 2023 | Minerva Math | AIME 2025 | Olympiad Bench |
|---|---|---|---|---|---|---|---|
| | | | *Post-training on MATH dataset* | | | | |
| Qwen-1.5B | w/o | - | 31.83 / 81.92 | 34.30 / 79.23 | 5.33 / 28.91 | 3.64 / 22.31 | 15.40 / 37.16 |
| | 2 | 2.05 | 67.73 / 87.85 | 53.28 / 86.21 | 14.15 / 34.02 | 6.15 / 29.54 | 23.11 / 37.82 |
| | 4 | 2.78 | 69.05 / 87.49 | 52.50 / 92.01 | 15.29 / 33.57 | 8.33 / 27.13 | 23.08 / 38.99 |
| | 8 | 4.67 | 69.34 / 86.05 | 51.64 / 83.96 | 14.60 / 32.63 | 7.18 / 32.24 | 22.77 / 36.69 |
| | 16 | 8.53 | 70.24 / 87.24 | 51.25 / 83.46 | 16.84 / 33.46 | 10.10 / 35.82 | 22.30 / 38.33 |
| Qwen-7B | w/o | - | 47.16 / 85.95 | 38.36 / 85.29 | 5.99 / 31.10 | 5.00 / 25.17 | 9.83 / 34.30 |
| | 2 | 2.43 | 74.41 / 89.25 | 63.83 / 89.58 | 21.53 / 37.72 | 11.67 / 33.05 | 26.04 / 41.34 |
| | 4 | 3.48 | 76.24 / 88.16 | 63.51 / 84.97 | 23.09 / 41.03 | 10.83 / 32.42 | 26.25 / 40.78 |
| | 8 | 5.48 | 75.12 / 89.53 | 64.38 / 88.63 | 22.24 / 35.94 | 12.71 / 35.85 | 26.25 / 40.52 |
| | 16 | 9.30 | 75.90 / 88.24 | 61.79 / 80.77 | 22.81 / 37.68 | 13.23 / 34.22 | 25.99 / 40.11 |
| | | | *Post-training on DAPO-Math-Sub dataset* | | | | |
| Qwen-1.5B | w/o | - | 31.83 / 81.92 | 34.30 / 79.23 | 5.33 / 28.91 | 3.64 / 22.31 | 15.40 / 37.16 |
| | 2 | 3.63 | 67.71 / 87.68 | 53.82 / 88.35 | 16.85 / 34.83 | 8.12 / 32.99 | 23.21 / 39.26 |
| | 4 | 4.90 | 69.14 / 87.78 | 54.69 / 86.88 | 17.53 / 35.74 | 8.43 / 36.18 | 23.30 / 39.00 |
| | 8 | 8.62 | 70.25 / 86.84 | 57.57 / 81.19 | 17.80 / 35.08 | 8.54 / 29.42 | 24.23 / 39.95 |
| | 16 | 13.30 | 70.66 / 87.03 | 56.56 / 85.53 | 18.00 / 34.16 | 9.58 / 32.31 | 24.55 / 39.19 |
| Qwen-7B | w/o | - | 47.16 / 85.95 | 38.36 / 85.29 | 5.99 / 31.10 | 5.00 / 25.17 | 9.83 / 34.30 |
| | 2 | 3.43 | 74.41 / 89.25 | 63.83 / 89.58 | 21.53 / 37.72 | 11.67 / 33.05 | 26.04 / 41.34 |
| | 4 | 5.39 | 76.24 / 88.16 | 63.51 / 84.97 | 23.09 / 41.03 | 10.83 / 32.42 | 26.25 / 40.78 |
| | 8 | 9.18 | 75.12 / 89.53 | 64.38 / 88.63 | 22.24 / 35.94 | 12.71 / 35.85 | 26.25 / 40.52 |
| | 16 | 17.68 | 75.90 / 88.24 | 61.79 / 80.77 | 22.81 / 37.68 | 13.23 / 34.22 | 25.99 / 40.11 |

# C The Use of Large Language Models (LLMs)

We used LLMs to polish the writing.

---

[4]It is worth noting that the batch size and learning rate used for 2-GRPO in this ablation differ from those employed in the main experiment.

