# OpenReview forum: "It Takes Two: Your GRPO Is Secretly DPO"
_ICLR.cc/2026/Conference — Submitted to ICLR 2026_

### Official Review · Reviewer_G2em · 2025-10-26

**Soundness:** 3
**Presentation:** 3
**Contribution:** 2
**Rating:** 6
**Confidence:** 3

**Summary:**

This paper reinterprets Group Relative Policy Optimization (GRPO) and Direct Preference Optimization (DPO) as a form of contrastive learning. Based on this insight, the authors propose 2-GRPO, an extremely lightweight variant that uses only two rollouts per prompt instead of large groups. , They show the 2-GRPO is able to preserve gradient estimation and the same optimization direction as standard GRPO. Through theoretical analysis, they prove that 2-GRPO implicitly normalizes advantages and maintains stability, and empirically show across multiple reasoning benchmarks that it matches the performance of 16-GRPO while largely cutting training time and computational cost.

**Strengths:**

The work theoretically and empirically showed that 2-GRPO can be compatible with large GRPO while reduce the rollout.

**Weaknesses:**

Although 2-GRPO achieves comparable performance to standard GRPO with substantially lower computational cost, it also presents several limitations. From Table 1 and figure 1,2, the overall performance is dropped. Especially, AIME drops up to 9.56% and AMC dropped up to 4.85%.

**Questions:**

1. Is there any way to reduce the drop in performance? Could we increase the number of epochs for 2-GRPO to match the performance of standard GRPO? If so, approximately how many additional epochs would be required?

2. How can the theory be generalized to continuous reward settings?

---

> ### Author Response · Authors · 2025-11-22
>
> > (W1) Although 2-GRPO achieves comparable performance to standard GRPO with substantially lower computational cost, it also presents several limitations. From Table 1 and figure 1,2, the overall performance is dropped. Especially, AIME drops up to 9.56% and AMC dropped up to 4.85%.
> >
> - We thank the reviewer for the valuable comments.
>     - We acknowledge that the peak performance of 2-GRPO slightly drops compared to the regular GRPO. However, compared to the improvement on the computational cost around $70\% \sim 85 \%$ — especially on the training time as well as the total rollouts traversed — the performance drop is totally acceptable.
>     Considering the **no-free lunch theorem [1]**, it is actually not reasonable to expect no drops observed at all.
>     Therefore, we kindly disagree that this shall be considered as a vital weakness.
> - “AIME dropped up to 9.56% and AMC dropped up to 4.85%.”
>     - We would like to clarify the setting of our experiments
>         - The experiments are under the Out-of-distribution (OOD) evaluation — the evaluation datasets are different from the training dataset. Therefore, the performance involves the OOD generalization determined by the distribution gap between the training and evaluation datasets. And it is expected that the performances across different evaluation datasets are non-consistent.
>         - Qwen-7B@DAPO shows $4.85\%$ drop of Mean@32 (i.e., Pass@1 across 32 times of evaluation) on AMC, but also shows $4.28\%$ improvement of Pass@32 on AMC
>         - Pass@32 is actually not a direct metric to evaluate the performance, but instead an associated metric to indicate the diversity of the model. In fact, DS-1.5B@MATH only drops $2\%$ of Mean@32 on AIME 2025, **which is actually lower than 1 question** (AIME 2025 only has 30 questions).
>
>
> > (Q1) Is there any way to reduce the drop in performance? Could we increase the number of epochs for 2-GRPO to match the performance of standard GRPO? If so, approximately how many additional epochs would be required?
> >
> - Considering the much lower computation cost consumed, it is a reasonable solution to reach stronger performance by increasing the training epochs.
> However, prolonging the training is not a trivial task [2], which requires careful setting of the KL penalty as well as the update of the reference models.
> - We have attempted simply prolonging the training and indeed observed improvements on several datasets. However, the improvement is not consistent due to the OOD setting. Therefore, we leave the in-depth study on it for future work.
>
> > (Q2) How can the theory be generalized to continuous reward settings?
> >
> - Our findings -  GRPO is actually doing contrastive learning or direct preference optimization (DPO) - can be directly generalized to continuous reward settings.
> However, the generalization is not trivial.
> Some sweet propositions and properties in our work might not hold due to the more complicated scenarios.
> Further study is necessary for this generalization.
> Therefore, due to the limited space in a conference paper, we leave the extension to continuous reward settings for future work.
> - Note that GRPO itself is not guaranteed to perform well in continuous reward settings due to the flaw of "Better than Average” in relative preference (one example is [3]). This is an issue of GRPO itself rather than the issue introduced by our work.
>
> [1]  David H. Wolpert and William G. Macready. No free lunch theorems for search. Vol. 10. No. 12. Technical Report SFI-TR-95-02-010, Santa Fe Institute, 1995.
>
> [2] Mingjie Liu, et al. "Prorl: Prolonged reinforcement learning expands reasoning boundaries in large language models." *arXiv preprint arXiv:2505.24864* (2025).
>
> [3] Anisha Garg and Ganesh Venkatesh. "The Peril of Preference: Why GRPO fails on Ordinal Rewards." *arXiv preprint arXiv:2511.04439* (2025).

---

### Official Review · Reviewer_W4J6 · 2025-11-03

**Soundness:** 2
**Presentation:** 3
**Contribution:** 1
**Rating:** 2
**Confidence:** 3

**Summary:**

The paper analyzes GRPO with group size of 2 and finds that it is able to achieve similar performance to group size of 16 on math datasets while using 70% wall clock time. Furthermore, a contrastive learning framework is used to analyze GRPO's advantage estimates under differing group sizes.

**Strengths:**

- A theoretical analysis under a constrastive learning framework for 2-GRPO is provided
- Established math datasets and benchmarks are used for post-training and evaluation

**Weaknesses:**

- My primary concern with the work is that while a 70% wall clock time improvement is significant, it appears that experimentally all that was done was changing the group size to 2
- I would need to see a rather elaborate experiments section to verify that G=2 is equally performant and significantly faster on a wider variety of post training tasks to fully believe the claim

**Questions:**

- Please address weaknesses

---

> ### Author Response · Authors · 2025-11-22
>
> > (W1) My primary concern with the work is that while a 70% wall clock time improvement is significant, it appears that experimentally all that was done was changing the group size to 2
> >
> - We kindly disagree that our work is simply experimental findings.
>     - First, the finding originates from the experimental observations. However, our work provides a thorough theoretical analysis on why this “shall not work” setting can work well. Therefore, it is inappropriate to claim that our work is simply experimental.
>     - Second, our work is very principled since what we need to do is change the group size to 2. A minimal change leading to a significant outcome is fundamental compared to complicated modifications, which shall be viewed as a strength rather than a weakness. (as stated in **Occam's razor principle**)
>
> > (W2) I would need to see a rather elaborate experiments section to verify that G=2 is equally performant and significantly faster on a wider variety of post training tasks to fully believe the claim
> >
> - Considering the **no-free lunch theorem [1]**, it is actually not reasonable to expect 2-GRPO is equally performant while it is significantly faster.
> In our experiments, we have demonstrated we can reach comparable performance with significant improvement on computational cost and training time (over $70\%$ faster).
> This already provides a strong outcome regarding the compute-performance trade-off.
> - We have covered the core tasks of GRPO. Combining our theoretical analysis, we believe this already provides strong justifications on 2-GRPO.
> - Considering the concerns from the reviewer, we are implementing extra experiments covering other post-training tasks where GRPO is applicable. However, it takes time. We will upload them as soon as possible.
>
> [1] Wolpert, David H., and William G. Macready. No free lunch theorems for search. Vol. 10. No. 12. Technical Report SFI-TR-95-02-010, Santa Fe Institute, 1995.

---

### Official Review · Reviewer_ZREz · 2025-11-05

**Soundness:** 3
**Presentation:** 2
**Contribution:** 1
**Rating:** 2
**Confidence:** 3

**Summary:**

The paper introduces a variant of the GRPO algorithm which takes group size to be 2. The paper proved that similar to DPO, GRPO with group size 2 can be considered as a contrastive loss. The paper then derives a few property of 2-GRPO, and perform experiments on standard math reasoning tasks and demonstrates the gain in efficiency with small performance degradation.

**Strengths:**

1. The observation that GRPO’s within-group normalization induces a signed “positive vs negative” split is a useful lens and does connect to generic contrastive forms.

2. The compute focus is valid given rollout cost dominates; the table shows large wall-clock savings.

**Weaknesses:**

1. The paper proves that both objectives admit gradients of a contrastive form (Def. 3.1 / Props. 3.2–3.3), but stops there. There is no mapping of corresponding terms in DPO to GRPO’s objective. This is just a nominal equivalence.

2. The assumption of no clipping is rather strong. I understand it's for the convenience of the derivation, but in reality it plays an important role on shaping the gradient distribution.

3. Honestly most of the observations / derivations in this paper is either quite well known or elementary (e.g., properties of Bernoulli random variables).

4. The results actually show that taking group size 4 induces both huge save in training time and no performance gap w.r.t. group size 8 or 16, that makes the choice of group size 2 less ideal.

5. This paper is less principled than many previous papers with adaptive group size. E.g., [1]

[1] Zhang, Ruiqi, et al. "SPEED-RL: Faster Training of Reasoning Models via Online Curriculum Learning." arXiv preprint arXiv:2506.09016 (2025).

**Questions:**

With group size 2, what is the percentage of prompts that have no gradients? How does this change along training?

---

> ### Author Response · Authors · 2025-11-22
>
> > (W1) The paper proves that both objectives admit gradients of a contrastive form (Def. 3.1 / Props. 3.2–3.3), but stops there. There is no mapping of corresponding terms in DPO to GRPO’s objective. This is just a nominal equivalence.
> >
>
>
>
> - Thank you for the suggestion. We have provided a more direct mapping in our revised version for easier connection (Eq (10) in Prop. 32 and Eq (11) in Prop. 3.3)
>
> > (W2) The assumption of no clipping is rather strong. I understand it's for the convenience of the derivation, but in reality it plays an important role on shaping the gradient distribution.
> >
> - When clipping is activated, the gradient of the GRPO objective w.r.t. $\theta$ for that token is zero.
> Therefore, we drop the clipping part for the simplicity of the equation, which will not impact our proposition.
> - However, we appreciate the concerns shared by the reviewer. Therefore, we write out the clipping part in our revised version for better understanding (In Sec 3.2, Line 165-202).
> We hope that the more detailed form can address the concerns from the reviewer.
>
> > (W3) Honestly most of the observations / derivations in this paper is either quite well known or elementary (e.g., properties of Bernoulli random variables).
> >
> - Our propositions/observations/derivations are for justifying why the **contrastive learning perspective** of GRPO can provide more insights compared to the **reward normalization perspective.**
> The contrastive learning perspective can explain why the two-rollout case of GRPO (2-GRPO) can still work effectively, which is viewed in appropriate from the normalization perspective.
>
>
>     ****We believe that using advance and complicated Math Tools is not the goal of Machine Learning research.
>     Instead, concretely and solidly explaining the findings in a mathematical language is far more important.
>     We think we can sufficiently carry out our findings and viewpoints using elementary math properties.
>     ****
>
>
> > (W4) The results actually show that taking group size 4 induces both huge save in training time and no performance gap w.r.t. group size 8 or 16, which makes the choice of group size 2 less ideal.
> >
> - We believe the reviewer might unintendedly miss our main result table (Table 1) or neglect the description in our ablation study (Table 2).
> - Table 2 shows an ablation study that ignores the variance reduction via a larger batch size.
> - In the main result table (Table 1), the performance of 2-GRPO is not worse than that of 4-GRPO in Table 2.
> - More importantly, we would like to provide empirical evaluation to justify that the contrastive learning nature of GRPO is more essential than performing reward normalization. Therefore, the empirical result of 2-GRPO is more valuable.
>
> > (W5) This paper is less principled than many previous papers with adaptive group size. E.g., [1]
> >
> - We kindly disagree that papers with adaptive group size are more principled, as whether the adaptive group size is a better solution remains an open question.
>
> In fact, we believe our work is more principled as our method is simpler, while we can reach comparable performance with significant efficiency gain. (as stated in **Occam's razor principle**)
>     - Note that the paper recommended by the reviewer still contains an issue — the results and conclusions may change —  as stated in their latest arXiv version. (https://arxiv.org/pdf/2506.09016)
>
> > (Q1) can be estimated by the reward
> >
> - The percentage will increase along the training as the LLM can answer with an increasing success rate.
> We did not conduct the exact counting during our experiments. However, the percentage can be directly reflected by the training rewards  (as shown in Figure 2) since we are under the RLVR setting (only True and False rewards).

---

### Official Review · Reviewer_Ako8 · 2025-11-05

**Soundness:** 3
**Presentation:** 3
**Contribution:** 3
**Rating:** 4
**Confidence:** 3

**Summary:**

The authors perform theoretical analysis and experiments in support of a variant of GRPO with group size 2.

**Strengths:**

(+) A mixture of theory and practice.

**Weaknesses:**

(-) I think you're missing a log in Eq. 2. Similarly, the expectation in Eq. 5 clashes with the finite-sample group.

(-) On line 272, I think you mean variance not bias. Also, I don't think you meant proportional on line 276. More broadly, it's not clear to me why Prop 4.1 implies that 2-GRPO performs normalization. I think there's also an extra nabla on line 286.

(-) The rationale for increasing Q to compensate for lower G in Sec. 4.2 seems to rely on a strong unstated assumption: equal variance across all prompts. I believe the same is true for Sec. 4.3 -- let me know if I've misunderstood.

**Questions:**

1. Why is the gradient zero outside of the clipping range as you write above Eq. 8?

2. Can you explain the justification for Prop. 3.2? It also seems like you might be missing a log in Eq. 10?

---

> ### Author Response · Authors · 2025-11-22
>
> > (W1) I think you're missing a log in Eq. 2. Similarly, the expectation in Eq. 5 clashes with the finite-sample group.
> >
>
> Thank you for pointing out all these. We have corrected Eq.2 and Eq.5 in our revised version.
>
> > (W2) On line 272, I think you mean variance not bias. Also, I don't think you meant proportional on line 276. More broadly, it's not clear to me why Prop 4.1 implies that 2-GRPO performs normalization. I think there's also an extra nabla on line 286.
> >
>
> On line 272, we do mean variance.
>
> - What we want to say in line 276 and Prop. 4.1 is that even though 2-GRPO only has 3 types of advantage (-1, 0, 1), these advantages still perform the weighting on positive and negative rollouts as regular GRPO. This is because the probability of appearing advantages still depends on the question’s success rate when we conduct stochastic optimization. This highlights that 2-GRPO can still weigh the positive/negative rollouts for different prompts based on their success rates.
> - You’re right. We mistakenly put an additional nabla on line 286. We have corrected it in our revised version (line 311).
>
> > (W3) The rationale for increasing Q to compensate for lower G in Sec. 4.2 seems to rely on a strong unstated assumption: equal variance across all prompts. I believe the same is true for Sec. 4.3 -- let me know if I've misunderstood.
> >
> - Sec. 4.2 indeed relies on the equal variance across all rollouts (i.e., the data to optimize the policy network). However, it is actually not a strong assumption.
>
>     It can be simply driven by the assumption — the prompts from the dataset are under an independent identical (i.i.d.) distribution  $\mathcal{S}$ — which is fair and held by most existing works analyzing machine learning algorithms.
>
>     Then $\boldsymbol{x} \sim \mathcal{D} = \mathcal{S}\times \mathcal{M}$ where $\mathcal{M}$ denotes the model randomness, shared by all prompts. Note that $\boldsymbol{x}_i$ is actually a random variable.
>
>     To better clarify the Def. 4.2 and Lemma 4.3, we introduce the concept of random variables in our revised version.
>
>     Note that even though we relax the i.i.d. assumption to independent non-identical, owing to the random sampling in mini-batch training, we can still get a similar conclusion since the distribution of training data in optimization (i.e., sampled rollouts from policy networks) shall be the same in mini-batch $B_1$ and mini-batch $B_2$. Therefore, the mean of the empirical variance remains the same.

---

> > ### Author Response · Authors · 2025-11-22
> >
> > > (Q1)  Why is the gradient zero outside of the clipping range as you write above Eq. 8?
> > >
> >
> > To discuss this problem, we need to take a look at PPO’s objective (Eq. 3).
> > The clipping is upper clipping $A \cdot \text{min}(f(\pi_\theta), 1+\epsilon)$ for positive advantage $A > 0$ and lower clipping $A\cdot \max(f(\pi_\theta), 1-\epsilon)$ for $A <0$, where $f$ denotes the importance sampling of $\pi_\theta$ for simplicity.
> > Therefore, when the value is clipped, the objective function becomes independent to $\theta$ and thus has zero gradient w.r.t. $\theta$ during backpropagation.
> >
> > Thus, we can drop those tokens from the sum notation in Eq.8 for simplicity. We add this no-clipping assumption mainly for the notation simplification and readability of the proof.
> >
> > To better clarify and avoid confusion, we rewrite the formula with importance sampling and clipping in the revised version (now Eq (9) on line 173)
> >
> > > (Q2) Can you explain the justification for Prop. 3.2? It also seems like you might be missing a log in Eq. 10?
> > >
> >
> > To better clarify Prop. 3.2, we have provide a revised version which writing out the details of each term (Line 186 in revised version).
> >
> > GRPO in practice split the rollouts into two groups — a positive group with positive advantages and a negative group with negative advantages — based on the group normalization on rewards.
> > This is equivalent to contrastive learning with $N$ positive samples and $M$ negative samples, where $N + M=G$. Therefore, the gradient of the GRPO objective is actually performing contrastive learning.
> >
> > Note that in the original definition (or the most general form) of Contrastive loss, the $\text{log}$ does not exist, as discussed in (Chopra et al., 2005; Hadsell et al., 2006). The choice of using $\text{log}$ in contrastive loss (such as InfoNCE (Oord et al., 2018)) is actually a design choice rather than a required component.
> > Therefore, the missing log here will not impact our conclusion that GRPO is doing contrastive learning.
> >
> > - Sumit Chopra, Raia Hadsell, and Yann LeCun. "Learning a similarity metric discriminatively, with application to face verification."  CVPR 2005.
> > - Raia Hadsell, Sumit Chopra, and Yann LeCun. "Dimensionality reduction by learning an invariant mapping." CVPR 2006.
> > - Aaron van den Oord, Yazhe Li, and Oriol Vinyals. "Representation learning with contrastive predictive coding." *arXiv preprint arXiv:1807.03748* (2018).

---

### Author Response · Authors · 2025-11-28

Dear reviewers,

We sincerely appreciate the time you have invested in reviewing our work.

If you have any further thoughts after reading our rebuttal, we would be grateful to hear them.

Best regards,

---

### Meta-Review · Area_Chair_5ahx · 2026-01-09

**Summary:**

This work revisits Group Relative Policy Optimization (GRPO) for post-training large language models and challenges the common belief that large group sizes are necessary for stable learning. By reframing GRPO as a form of contrastive learning, the authors clain to establish a connection to Direct Preference Optimization (DPO) and show that GRPO remains theoretically sound even in the minimal two-rollout setting. Both theoretical analysis and empirical results are reported towards justifying simplified 2-GRPO variant can achieve performance comparable to standard GRPO while significantly reducing computational overhead.

**Reviewer Concerns:**

- Limited depth in the theoretical connection to prior objectives: While the paper shows that GRPO and DPO admit gradients of a contrastive form, reviewers felt the analysis stops short of establishing a meaningful correspondence between the two objectives. The equivalence remains largely formal, without a clear mapping of terms or deeper conceptual insight.

- Strong and unrealistic modeling assumptions: The theoretical analysis relies on assumptions such as the absence of clipping and uniform variance across prompts, which are unlikely to hold in practical training settings and may substantially affect gradient behavior.

- Several derivations and observations are viewed as elementary or well understood in prior work, limiting the perceived novelty and technical depth of the contribution.

- Limited experimental validation: The empirical evaluation primarily varies group size and does not sufficiently demonstrate that the proposed approach generalizes across a broader range of post-training tasks, model scales, or training regimes. Reviewers also note that the approach appears less principled than prior work on adaptive or curriculum-based group sizing, and the paper does not adequately position itself relative to this literature.

**Reviewer Scores:**

None responded to the rebuttal, so I have no idea.

---

### Decision · Program_Chairs · 2026-01-26

Reject